# Ethanol Exposure Increases Oxygen Consumption by Developing Cerebral Arteries in a Trimester-, Concentration- and Sex-Dependent Manner

**DOI:** 10.3390/biom15111566

**Published:** 2025-11-07

**Authors:** Shiwani Thapa, Rika M. Morales, Heather S. Smallwood, Anna N. Bukiya

**Affiliations:** 1Department of Pharmacology, Addiction Science and Toxicology, College of Medicine, The University of Tennessee Health Science Center, Memphis, TN 38103, USA; sthapa8@uthsc.edu (S.T.); rmorale6@uthsc.edu (R.M.M.); 2Department of Pediatrics, College of Medicine, The University of Tennessee Health Science Center, Memphis, TN 38103, USA; hsmallwo@uthsc.edu

**Keywords:** maternal drinking, alcohol in utero, mitochondria, respiratory chain, Seahorse

## Abstract

Alcohol (ethanol; EtOH) intake affects one in ten pregnancies in the United States and is a leading cause of developmental defects collectively known as fetal alcohol spectrum disorders (FASDs). Cerebral circulation is a critical target of prenatal ethanol exposure (PEE), yet the target(s) involved remain poorly understood. In adult cerebral circulation, mitochondrial function is essential in regulating smooth muscle contractility, suggesting mitochondria as a potential target of alcohol in the developing cerebral arteries. In this study, pregnant C57BL/6J mice were administered ethanol (3, 4.5, 6, or 7 g/kg) during either the second trimester equivalent of human pregnancy (gestational days 9–19), or the third trimester equivalent during postnatal days 1–10. Maternal and progeny blood ethanol concentrations, progeny brain weight, cerebral artery oxygen consumption, and corticosterone levels were measured. At lower ethanol concentrations (3 g and 4.5 g/kg), no significant alterations in fetal cerebral artery mitochondrial function were detected. In contrast, heavy maternal ethanol exposure (6 g/kg) significantly increased mitochondrial respiratory parameters in developing cerebral arteries during the third trimester equivalent of human pregnancy. Sex-specific dimorphism was also observed at this developmental stage. Corticosterone was not elevated in fetuses and pups. In summary, our findings demonstrate developmental stage- and sex-dependent vulnerabilities of cerebrovascular oxygen consumption to ethanol exposure.

## 1. Introduction

Prenatal ethanol exposure (PEE) is a leading preventable cause of developmental disorders worldwide [1,2,3,4,5,6]. Globally, about 10% of pregnant women partake in consuming alcohol [4,7]. Depending on the stage of gestation and dose, ethanol targets multiple fetal organs and systems, resulting in a range of craniofacial, neurodevelopmental, and behavioral abnormalities collectively termed as fetal alcohol spectrum disorders (FASDs) [1,5]. FASD has a global prevalence of ~0.77%, with rates in certain communities reaching 30% [1,2,3,5,6]. Despite its high prevalence, there is no known cure for FASD. Therefore, understanding the mechanisms underlying FASD is clinically important for developing preventive strategies and mitigating long-term outcomes.

The fetal brain is especially vulnerable to ethanol, with decreased brain volume, long-term structural and functional impairments being reported [8,9,10,11]. Cerebral circulation is a key regulator of overall brain development as it facilitates progenitor cell migration, oxygen and nutrient supply, and waste removal [12,13,14,15,16]. Cerebrovascular development begins around 24 days of human gestation with the formation of the Circle of Willis and its four major cerebral arteries (anterior, middle, posterior, and basilar), complete by the end of the first trimester of pregnancy [12,14]. Fetal cerebral circulation has been advanced as a critical target of PEE [11,17,18,19,20,21,22,23]. PEE has been shown to impair cerebral vascularization and reduce microvessel density [9,24,25]. A proteomics study from our lab on non-human primate cerebral arteries showed that maternal binge ethanol exposure during mid-pregnancy alters protein networks [18]. Specifically, ethanol exposure in utero increases the mitochondrial proteome [18]. Mitochondria consume up to 98% of the total oxygen supplied to tissue and are central to energy production for different cellular processes essential for cerebral vasculature formation and function [26,27,28,29]. Thus, cerebrovascular mitochondria may be critical for maintaining overall brain development. Mitochondrial dysfunction can impair vascular tone and angiogenesis [30], exacerbating long-term vascular and metabolic consequences of PEE [21,24,31,32]. Ethanol metabolism produces acetaldehyde and reactive oxygen species, both of which contribute to oxidative stress, mitochondrial damage, and altered oxygen availability in developing tissues [33,34,35]. Mitochondrial dysfunction has been documented in neuronal and glial models of PEE [36,37,38,39,40,41]; however, the impact of ethanol on mitochondria within the developing cerebrovasculature remains poorly understood.

While scarce evidence indicates that cerebral vasculature mitochondria may be a target of PEE [42], the effect of early ethanol exposure on mitochondrial function within developing cerebral arteries remains unknown. In this study, we investigate the effects of ethanol exposure on cerebral artery oxygen consumption and related parameters of mitochondrial respiration in a C57BL/6J mouse line. Our findings reveal a distinct developmental window of vulnerability and highlight sex-dependent differences, providing new insight into how ethanol disrupts cerebrovascular mitochondrial function during brain development.

## 2. Materials and Methods

### 2.1. Animal Subjects and Experimental Groups

Eight- to twelve-week-old male and nulliparous female C57BL/6J mice were purchased from Jackson Laboratory and acclimated for at least 3 days upon arrival to UTHSC. Time-matched single pair breeding was maintained following the Jackson Laboratory protocol [43]. To confirm mating, the female mice were checked for vaginal/copulation plugs the following morning. Pregnancy was validated by measuring female weight on GD 1 and 8, ensuring a weight gain of at least 1.75 g [44,45]. After confirming pregnancy, a multiple ethanol exposure model was applied to four groups of pregnant female mice at different gestational and postpartum days, corresponding to the second or third trimester of human pregnancy, respectively. The pregnant females were randomly assigned to one of the four groups: control versus ethanol exposure (GD 9–19, second trimester equivalent of human pregnancy) and control versus ethanol exposure (PD 1–10, third trimester equivalent of human pregnancy). Ethanol was administered to dams via intragastric gavage using 40% ethanol (prepared from 190 proof 95% ethanol) at 12 h intervals for 10 consecutive days [17,46]. During the third-trimester-equivalent group, ethanol was administered via intragastric gavage to nursing dams; thus, pups were exposed to alcohol through ingestion of maternal milk. Different ethanol concentrations were used for exposure: 3 g/kg, 4.5 g/kg, 6 g/kg, and 7 g/kg of the dam’s weight. The control groups received gavages of distilled water that were administered on the same days as ethanol gavages to the ethanol exposure groups.

### 2.2. Measurement of Progeny Brain and Body Weight

Following completion of the control versus ethanol exposure regimen during the second trimester equivalent of human pregnancy, fetuses were harvested on GD 19, and their brains were weighed by an experimenter who was blind to the identity of the experimental group. Similarly, following completion of the control versus ethanol exposure regimen during the third trimester equivalent of human pregnancy, pup brains were harvested on PD 11, and their brain weights were measured by a blinded experimenter. Pup body weights were measured on PD 10, one day prior to subsequent experiments, to minimize handling-related stress on the day of the experiment.

### 2.3. Measurement of Blood Ethanol Levels

Blood was collected from dams two hours after the first intragastric gavage using a tail nick. For the pups, the blood was collected during thoracotomy under deep anesthesia with isoflurane, 12 h post the last intragastric gavage. Pup blood was not collected earlier to minimize handling-related stress. The blood samples were centrifuged at 14,000 rpm at 4 °C for 30 min. The ethanol levels in the supernatant were measured using the Nicotinamide Adenine Dinucleotide–Ethanol Dehydrogenase Reagent kit following the manufacturer’s instructions (N7160, Sigma-Aldrich, St. Louis, MO, USA).

### 2.4. Oxygen Consumption and Analysis of Mitochondrial Respiration Parameters in Developing Cerebral Arteries

To assess gross mitochondrial function, progeny cerebral arteries were harvested 12 h after the last intragastric gavage dose for all groups. Pregnant dams, postpartum dams, and progeny were humanely euthanized by decapitation under deep anesthesia with isoflurane. For the GD 9–19 groups, cerebral arteries of six fetuses from the same dam were pooled together to obtain enough tissue for a single datapoint (“*n*”). For the PD 1–10 groups, cerebral arteries of three pups from the same dam were pooled together. These pooled cerebral arteries were loaded into individual wells of the microplate and underwent the Agilent Seahorse XF Cell Mito Stress Test to measure oxygen consumption rate (OCR) as an indicator of mitochondrial respiration. The test was performed using the Agilent Seahorse XFe96 Analyzer (Seahorse Bioscience, N. Billerica, MA, USA). For this assay, four different modulators of mitochondrial respiratory chain were used: 2 μM oligomycin (ATP synthase blocker), 1 μM carbonyl cyanide 4-(trifluoromethoxy) phenylhydrazone (FCCP, mitochondrial uncoupler), 0.5 μM rotenone, and 0.5 μM antimycin (blockers of respiratory chain proteins I and III, respectively; Appendix A). The protocol was optimized following the manufacturer’s instructions. The assay protocol consisted of three cycles of baseline measurements followed by five cycles of oligomycin and FCCP drug treatment, and finally three cycles of rotenone/antimycin mixture. Data was analyzed using the Seahorse Wave Software (Wave 2.6). Only mitochondria-associated OCR values are plotted, as non-mitochondrial respiration rates (usually observed in presence of rotenone/antimycin mixture) were subtracted from oxygen consumption values (see Appendix A). Any data where the baseline OCR skewed by more than 50% from the initial value was excluded from further analysis. The equations used to calculate each mitochondrial respiration parameter are summarized in Table 1.

### 2.5. Quantification of Total DNA Content in Progeny Cerebral Arteries

The cerebral arteries collected from the fetuses and pups (as described above) were immediately frozen at −80 °C after undergoing the Agilent Seahorse Mitochondria Stress Test assay. Total DNA content was measured using the Cyquant Cell Proliferation Assay following the manufacturer’s instructions (C7026, ThermoFisher Scientific, Eugene, OR, USA). This assay was used to normalize the OCR values to the total DNA content of each well’s sample.

### 2.6. Measurement of Corticosterone Levels

Blood was collected after thoracotomy from both dams and progeny under deep anesthesia with isoflurane, 12 h after the last intragastric gavage dose. The blood corticosterone levels were quantified at the same time point used for mitochondrial respiration measurements. The blood was centrifuged at 14,000 rpm at 4 °C for 30 min, and the supernatant was immediately collected and frozen at −80 °C. The corticosterone levels in the supernatant were measured using the Corticosterone ELISA Kit following the manufacturer’s instructions (EIACORT, Invitrogen, Frederick, MD, USA).

### 2.7. RNA Isolation and Quantitative PCR (qPCR) Analysis of Sry (Sex-Determining Region Y) Gene Expression

Immediately after euthanasia, the pups were inspected for the presence of apparent mammary glands to separate females from males. Tissue for qPCR analysis was collected immediately following visual sex separation. Male versus female pup livers were collected to ensure that the total amount of liver sample from each pup reached 5 mg. The tissue samples were immediately frozen at −80 °C for subsequent RNA isolation. Total RNA extraction from the pup livers was performed using the RNeasy Plus Micro Kit (74004, Qiagen, Hilden, Germany) following the manufacturer’s instructions. RNA concentration was determined using a NanoDrop One Microvolume UV-Vis Spectrophotometer (ThermoFisher Scientific, Madison, WI, USA), and RNA was stored at −80 °C. For cDNA synthesis, the High-Capacity cDNA Reverse Transcription Kit (4368814, Applied Biosystems, Vilnius, Lithunia) was used in a ProFlex PCR system (Applied Biosystems, Marsiling Industrial Estate Rd 3, Singapore). qPCR was conducted on a LightCycler 480 (Roche, Indianapolis, IN, USA) using the TaqMan Fast Advanced Master Mix and TaqMan gene expression assays. TaqMan probes used included *Sry* (Mm00441712_s1, ThermoFisher Scientific, Pleasanton, CA, USA) and *Gapdh* (Mm99999915_g1, ThermoFisher Scientific, Pleasanton, CA, USA). All the samples were run in triplicate for both target (*Sry*) and reference (*Gapdh*) genes following the manufacturer’s protocols. The gene expression levels were quantified using the 2^−ΔCt^ method as per the published guidelines and validation in our lab [47,48].

### 2.8. Chemicals Reagents

Ethanol (190 proof ethanol) was purchased from Thermo Scientific. To ensure minimal contact of ethanol with humid air, ethanol was aliquoted into smaller vials upon arrival and stored according to the manufacturer’s instructions. Oligomycin, FCCP, rotenone, and antimycin were purchased from Sigma Aldrich (St. Louis, MO, USA).

### 2.9. Statistical Analysis

Data plotting and fitting were performed using Origin 2022 (OriginLab Corp 2022). Outliers were detected and excluded using Origin 2022’s built-in function, which automatically applies either Grubbs or Dixon’s Q-test depending on the dataset. Statistical analysis was performed using the online tool, Statistics Kingdom. Statistical comparisons were conducted using two-tailed Welch’s *t*-test, one-way ANOVA with Tukey’s Honestly Significant Difference (HSD) test, and two-way ANOVA test. The specific statistical methods used are detailed in the figure legends. The difference between groups with the *p*-value of less than 0.05 was considered statistically significant. Data are presented as mean ± standard error. “*n*” values represent independent biological replicates corresponding to individual dams. In a few cases, a maximum of two data points were obtained from the same dam without compromising biological independence. For sex-specific analyses, no more than one male and one female datapoint per dam were analyzed when available.

## 3. Results

### 3.1. Developmental Stage- and Concentration-Specific Effect of Ethanol on Mitochondrial Respiratory Parameters of Developing Cerebral Arteries

We first tested the hypothesis that ethanol exposure alters cerebral artery oxygen consumption and mitochondrial respiration parameters in a trimester-specific and concentration-dependent manner. Pregnant C57BL/6J mice were exposed to 3 g/kg, 4.5 g/kg, or 6 g/kg ethanol twice a day via oral gavage from GD 9 to 19, corresponding to the second trimester of human pregnancy ([49,50,51,52], Figure 1A). The control group received gavages with distilled water. Volume of control gavages matched volume of ethanol-containing ones. The maternal blood ethanol levels, measured two hours after first ethanol exposure, were 92 ± 2 mg/dL, 94 ± 5 mg/dL, and 105 ± 4 mg/dL for the 3 g/kg, 4.5 g/kg, and 6 g/kg groups, respectively (Figure 1B). Exposure to 6 g/kg ethanol was lethal, with more than 50% maternal mortality during the ethanol exposure. Fetal mortality among dams exposed to 3 and 4.5 g/kg ethanol remained low, and litter sizes were not significantly different from those of control groups (Appendix A). The fetal blood ethanol levels were not significantly different between the two remaining ethanol exposure groups (3 g/kg vs. 4.5 g/kg ethanol; Figure 1C). The fetuses were delivered by cesarean section on GD 19, and their brains were harvested to record brain weights. Fetal cerebral arteries were dissected out to assess oxygen consumption using the Seahorse Mito Stress Test assay and to define mitochondrial respiration parameters.

Fetal brain weights remained unaffected in the 3 g/kg maternal ethanol group when compared to the controls (Figure 1D). However, basal OCR (basal respiration), maximal OCR, and ATP production-associated oxygen consumption exhibited a trend towards increased values compared to the control group (*p* = 0.08–0.1, Figure 1F). While fetal brain weights were significantly reduced in the 4.5 g/kg maternal ethanol group compared to the 3 g/kg group (*p* < 0.05) and showed a trend toward reduction compared to the control group (*p* = 0.16, Figure 1D), there were no noticeable trends or significant changes in oxygen consumption and mitochondrial respiratory parameters whether comparing 4.5 g/kg maternal ethanol group to 3 g/kg or control counterparts (Figure 1E,F).

The same multiple binge ethanol exposure model was used from PD 1 to 10, equivalent to the third trimester of human pregnancy ([49,50,51,52], Figure 2A). During this developmental window, dams were less sensitive to ethanol. This resulted in a lethal dose of 7 g/kg of ethanol. The maternal blood ethanol levels averaged 83 ± 2 mg/dL, 91 ± 4 mg/dL, 130 ± 7 mg/dL, and 145 ± 10 mg/dL for the 3 g/kg, 4.5 g/kg, 6 g/kg, and 7 g/kg ethanol groups, respectively (Figure 2B). The offspring blood ethanol levels were also elevated in the 6 g/kg maternal ethanol exposure group (67 ± 3 mg/dL) compared to the 4.5 g/kg group (*p* < 0.05, Figure 2C). Pup brain weights of all the ethanol-exposed groups were significantly decreased compared to the controls (*p* < 0.05, Figure 2D). Out of all the mitochondrial respiration parameters, spare respiratory capacity was only increased in the 3 g/kg maternal ethanol exposure group compared to the controls (*p* < 0.05, Figure 2E,F). None of the mitochondrial respiration parameters were changed by 4.5 g/kg maternal ethanol. In the 6 g/kg maternal ethanol group, basal respiration, maximal respiration, and ATP production-associated oxygen consumption were significantly elevated compared to the corresponding control (*p* < 0.05, Figure 2E,F). In addition, spared respiratory capacity exhibited a trend towards increased value in the ethanol group compared to the controls (*p* = 0.1, Figure 2F). These findings suggest that cerebral artery mitochondrial function is particularly sensitive to high-dose ethanol exposure during the third trimester equivalent of human pregnancy.

### 3.2. Sex-Specific Effects of Ethanol on Mitochondrial Respiration Parameters in Developing Cerebral Arteries During Third Trimester Equivalent of Human Pregnancy

We next determined whether progeny sex influences the vulnerability of cerebral artery oxygen consumption and mitochondrial respiration parameters to maternal ethanol exposure. Male and female pups were collected from litters of dams receiving 6 g/kg ethanol every 12 h during postpartum days 1 through 10 (third trimester equivalent of human pregnancy). The female pups were visually identified by presence of noticeable nipples of the mammary glands. Accuracy of sex determination was further confirmed by qPCR analysis of the sex-determining region Y (*Sry*) gene transcript in liver samples. qPCR data revealed accurate sex-based separation (Appendix A). Blood ethanol levels measured 12 h post final ethanol exposure (6 g/kg EtOH) showed no significant differences between the male and female pups (*p* = 0.66 by Welch’s *t*-test, Figure 3A). The brain and body weights were significantly decreased by ethanol exposure in both sexes compared to the controls (*p* < 0.05, two-way ANOVA, factor: ethanol exposure [#]), with no significant sex-based differences within ethanol-exposed groups (Figure 3B,C). The Mitochondria Stress Test assay revealed a main effect of sex [$] in maximal respiration rates (*p*[$] < 0.05 by two-way ANOVA), with males exhibiting higher values than females, and a trend toward higher proton leak and spared respiratory capacity in males (*p*[$] = 0.07 and *p*[$] = 0.1, respectively, by two-way ANOVA) (Figure 3D–F). In addition, there was a trend for a main effect of ethanol exposure [#] on ATP production–associated oxygen consumption (*p*[#] = 0.09), suggesting that ethanol exposure may increase ATP-linked respiration regardless of sex (Figure 3F). These findings highlight that ethanol exposure alters cerebral artery oxygen consumption and mitochondrial respiration parameters in a sex-specific manner during the third trimester equivalent to human pregnancy.

### 3.3. Corticosterone Levels Are Not Increased in Progeny of Ethanol-Exposed Dams

We then investigated whether increased corticosterone, a key stress hormone, contributes to ethanol-induced changes in cerebral artery oxygen consumption and mitochondrial respiration parameters. Serum corticosterone levels were measured in progenies from control versus ethanol-exposed groups using both second- and third-trimester-equivalent paradigms. In the dataset from the second trimester equivalent, fetal corticosterone levels were significantly decreased in the ethanol-exposed group (*p* < 0.05, Welch’s *t*-test, Figure 4A). In the dataset from the third trimester equivalent, offspring exposed to 6 g/kg ethanol of both sexes had reduced corticosterone (*p* < 0.05 by Welch’s *t*-test within each sex, *p* < 0.05, two-way ANOVA, factor: treatment [#]) (Figure 4B). Therefore, ethanol exposure-induced mitochondrial dysfunction is not accompanied by increased corticosterone levels.

## 4. Discussion

In this study, we performed ex vivo biochemical assays to determine the effects of PEE on cerebral artery oxygen consumption rate and to define parameters of mitochondrial respiration as indicators of mitochondrial function. Our findings demonstrate, for the first time, that PEE produces trimester-, dose-, and sex-dependent alterations in mitochondrial respiratory parameters. We also show that these changes are unlikely to be triggered by increased corticosterone in the progeny.

PEE is a well-known teratogenic insult that most severely targets the developing brain. PEE leads to lifelong physical, cognitive, and behavioral impairments collectively termed as FASD [1,5]. While the placenta regulates the maternal–fetal exchange of oxygen and ethanol, our investigation specifically targeted fetal cerebral arteries to examine local cerebrovascular mechanisms rather than systemic transport processes. Proper cerebral circulation is critical for brain development, ensuring oxygen and nutrient delivery, waste removal, and support of neuronal survival [12,13,14,15,16,53]. Many studies across multiple species, including sheep, mice, and non-human primates, have shown that maternal ethanol intake disrupts fetal cerebral blood flow and vascular development [11,17,18,19,20,21,22,23]. Cerebrovascular disruption has been proposed to be a critical contributor to the neuropathology of FASD [21,42]. Work from our laboratory revealed that repeated ethanol exposure during the second trimester equivalent of human pregnancy in non-human primates profoundly altered the fetal basilar artery proteome near pregnancy term, including cytoskeletal protein downregulation and mitochondrial protein upregulation [18]. Mitochondria are involved in maintaining different cerebrovascular processes such as arterial tone, vasoactive responses, and apoptosis [26,54,55,56]. While previous studies have reported PEE-induced mitochondrial alterations in neurons and glia, data on cerebrovascular mitochondria remain scarce. Mitochondrial dysfunction can impair vascular tone and angiogenesis [30], thus exacerbating long-term vascular and metabolic consequences of PEE [21,24,31,32]. PEE has also been reported to disrupt mitochondrial bioenergetics, dynamics, and signaling in a cell-type and stage-specific manner [42,57,58]. We hypothesized that cerebral artery mitochondrial respiration is sensitive to ethanol exposure in a trimester- and concentration-dependent manner. To test this, we performed a trimester-specific ethanol exposure paradigm in pregnant mice with varying toxicologically relevant ethanol concentrations (Figure 1A and Figure 2A). In our model, 3 g/kg ethanol gavage rendered around 80 mg/dL (0.08%) blood ethanol, which reflects the legal intoxication limit for driving a motor vehicle in the United States [59]. In addition, 4.5 g/kg rendered to ~90 mg/dL (0.12%) and 6 g/kg rendered ~120 mg/dL (0.16%) blood ethanol levels. These blood alcohol levels represent a range between moderate and heavy binge drinking patterns in humans [59,60,61,62,63]. Interestingly, while the maternal blood ethanol levels increased proportionally with dose, the minimum blood ethanol levels in the pups after repeated gavage remained consistently low, reflecting placental and metabolic modulation of ethanol transfer (Figure 1B,C and Figure 2B,C). Moreover, the pup (third trimester equivalent) blood ethanol concentrations measured two hours after maternal gavage remained low and were comparable to those measured 12 h after the last gavage (Appendix A). This observation is consistent with previous reports demonstrating that maternal breast can act as a partial filter, limiting ethanol transmission to offspring [64]. This observation also suggests that it may not be the ethanol molecule itself that triggers mitochondrial alterations, but rather an ethanol metabolite or a signaling molecule acting in addition to ethanol. Furthermore, the second-trimester-equivalent period appeared more vulnerable to our regimen of ethanol exposure than the third, as maternal lethality during the second trimester equivalent was observed with relatively low blood ethanol levels (≈100 mg/dL) as opposed to 140 mg/dL in the third-trimester-equivalent paradigm. These blood levels are below the blood ethanol that is considered lethal in mammals [59]; thus, pregnancy and lactation may be particularly vulnerable periods for ethanol toxicity and associated metabolic changes.

During the second trimester equivalent, both brain weights and mitochondrial function remained resilient to ethanol concentrations (Figure 1D–F). This suggests that the mitochondrial function of cerebral arteries may be more resilient earlier in development and is able to maintain the energy demand even upon toxic environmental insult. However, this resilience stage may not imply overall protection, as subtle molecular or proteomic changes may occur within the cerebrovascular system long-term, as there are statistical trends present towards increased basal respiration, maximal respiration, and ATP production-associated oxygen consumption after exposure to low-dose maternal ethanol (3 g/kg) (Figure 1F).

In contrast to the relative resilience of second-trimester-equivalent mouse fetuses to ethanol, during the third trimester equivalent, ethanol exposure significantly reduced brain weight across all the maternal ethanol exposure levels, indicating impaired brain growth (Figure 2D). Surprisingly, mitochondrial oxygen consumption rates were upregulated at the highest maternal ethanol dose of 6 g/kg with significant increases in basal respiration, maximal respiration, and ATP production-associated oxygen consumption, and a trend in spare respiratory capacity (Figure 2E,F). Despite the growth restriction, the upregulation of mitochondrial respiration and energy production may represent a compensatory response aimed at supporting neuronal survival. Similar compensatory mitochondrial upregulation has been reported in response to a lack of nutrient supply [65]. Ethanol-induced increases in oxygen consumption have also been associated with elevated reactive oxygen species generation and oxidative stress, reflecting a heightened mitochondrial workload under toxic conditions [65,66,67,68]. Consistent with these findings, the enhanced OCR observed in our study may indicate increased oxidative demand or mitochondrial strain in developing cerebral arteries. In a recent in vivo study, chronic PEE showed reduced levels of oxygen saturation in cerebral arteries, indicating increased oxygen demand due to changes in cerebral artery density [69]. This could imply that impaired vascular function causes nutrient crises in the developing brain and hence triggers compensatory mitochondrial function upregulation. This pattern may indicate an adaptive response of cerebrovascular mitochondria to metabolic stress. While increased oxygen consumption can also arise from mitochondrial uncoupling, the absence of robust elevation in proton leak (Figure 1F and Figure 2F) in our data suggests this was not the case. However, without direct measurements of mitochondrial membrane potential, ATP yield, reactive oxygen species generation, or vascular reactivity, the mechanistic interpretation remains speculative. Future studies employing imaging and functional assays will be necessary to determine whether the elevated oxygen consumption reflects a compensatory mitochondrial response or an early indicator of vascular dysfunction. Our findings also highlight that mitochondrial alterations occurred even when pup blood ethanol levels were lower than maternal levels, thus underscoring the heightened vulnerability of the developing cerebrovascular system to ethanol-triggered pathology.

Next, we investigated sex-dependent vulnerability of developing mouse pups to ethanol. Blood ethanol levels were not significantly different between the two sexes (Figure 3A). Growth parameters, including body and brain weights, were significantly reduced by ethanol regardless of sex, suggesting a broad toxic effect on developmental growth (Figure 3B,C). However, the oxygen consumption assay (Figure 3D,E) revealed that ethanol may have the opposite effects on several parameters of mitochondrial respiration in male versus female pups. Specifically, for maximal respiration, there is a statistically significant influence of sex on the assay outcome: while maximal respiration is increased by ethanol in males, it is decreased in females (Figure 3F). In spare respiratory capacity, there is a statistical trend towards sex-specific outcomes of ethanol exposure. This suggests intrinsic sex-dependent mitochondrial sensitivity, consistent with previous reports of male-biased vulnerability to developmental insults [70,71,72,73]. Males may have lower antioxidant levels or increased generation of reactive oxygen species, whereas estrogen signaling in females may confer mitochondrial function protection. Studies have shown an early perinatal period of mini puberty where female offspring have a burst of estradiol levels [74]. Furthermore, the observed sex-specific differences in mitochondrial parameters may be influenced not only by transient estradiol surges in females but also by testosterone-dependent modulation of mitochondrial bioenergetics in males. Sex hormones have been shown to affect mitochondrial biogenesis, redox balance, and calcium handling, which may contribute to differential susceptibility to metabolic stress between male and female offspring [75,76,77]. In contrast to sex-sensitive maximal respiration and spare respiratory capacity, proton leak and ATP production-associated oxygen consumption are decreased and increased, respectively, disregarding pup sex. The trend of increased ATP production-associated oxygen consumption, regardless of sex, may indicate that ethanol exposure increases mitochondrial ATP turnover, reflecting a generalized stress response. Of note, the overall effects of high-dose ethanol (6 g/kg EtOH) were less pronounced when data were collected separately by sex compared to the combined sexes (Figure 2E,F vs. Figure 3D–F). This attenuation may reflect the influence of intrinsic, sex-dependent factors released by either male or female offspring that modulate mitochondrial oxygen consumption rates. Interestingly, the overall survival of female pups was about 23% lower than males in the litters of ethanol-exposed dams (Appendix A). This effect was not noted in the control group. This suggests that female offspring may be more vulnerable to survival challenges. These findings underscore the importance of sex as a biological variable in FASD research, suggesting that interventions may need to be sex specific.

Previous studies have implied that glucocorticoids may exacerbate neurodevelopmental injury by promoting oxidative stress and impairing mitochondrial signaling [78,79]. Thus, we investigated whether the increase in stress hormone, corticosterone, contributed to ethanol-induced mitochondrial changes in developing cerebral arteries. Corticosterone levels were decreased, or tended to decrease, in offspring of ethanol-exposed dams in both trimesters compared to the controls, regardless of progeny sex (Figure 4A,B). These results did not correlate with the observed mitochondrial dysfunction, thus implying that an increase in corticosterone does not mediate ethanol’s effect on cerebrovascular mitochondrial function. Rather, it could be ethanol or triggered metabolic cascades and toxicity. However, as corticosterone levels were measured at a single time point after completion of ethanol gavages, our results represent a limited snapshot of hypothalamic–pituitary–adrenal axis activity. Therefore, while our data indicates no sustained increase in corticosterone at this time point, transient fluctuations earlier after ethanol exposure cannot be excluded. Future studies should incorporate time-course profiling of stress hormones to confirm whether dynamic hypothalamic–pituitary–adrenal responses contribute to the observed mitochondrial effects. In addition to glucocorticoids, other endocrine factors may also interact with immune and glial signaling pathways to influence cerebrovascular responses [80,81]. Microglia are highly sensitive to both stress hormones and gonadal steroids, and their activation can alter mitochondrial metabolism and vascular integrity during development [82,83,84,85,86]. Considering these interactions may provide a more comprehensive understanding of how ethanol exposure disrupts developmental neurovascular and metabolic homeostasis.

We must recognize, however, the limitations of our study that are inherent to experimental design. Control groups paired with ethanol exposures showed slight variability in measured oxygen consumption and quantified parameters of mitochondrial respiration (Figure 1F and Figure 2F). This points out the high sensitivity of developing cerebral arteries to environmental influences outside of our control. It should also be noted that findings described as trends (0.05 < *p* ≤ 0.1) are not considered biologically significant within the present dataset and should be interpreted cautiously given inherent variability. Validation in larger cohorts is required to substantiate these observations. Moreover, brain weight, although widely used as a developmental measure, may not be the most sensitive indicator of ethanol-induced neurovascular alterations, as changes in the overall brain weight may not reflect the full extent of neuronal damage [87,88,89,90,91,92,93,94]. While we assessed corticosterone as a marker of stress, additional neuroendocrine mediators may also play a role.

Despite limitations, our study used a model of multiple binge-pattern ethanol exposure with physiologically meaningful blood ethanol levels and directly examined cerebrovascular mitochondria function, an understudied target in FASD. Our approach provides new insights into stage-, dose-, and sex-specific vulnerabilities not captured before in neurodevelopmental models. Future studies should focus on assessing the roles of mitochondrial respiratory chain protein-specific function, quantifying ethanol metabolites, and exploring whether interventions targeting mitochondrial stability can mitigate cerebrovascular dysfunction. Longitudinal studies are also needed to establish whether these early mitochondrial alterations persist in later development and contribute to vascular and neurodevelopmental deficits characteristic of FASD.

## 5. Conclusions

In conclusion, our findings show how PEE disrupts mitochondrial respiration in developing cerebral arteries. To the best of our knowledge, this is the first study to demonstrate that early ethanol exposure disrupts cerebrovascular bioenergetics in a trimester-, dose-, and sex-dependent manner. Changes in mitochondrial respiratory parameters in the third trimester equivalent to human pregnancy may represent an adaptive response to metabolic stress or a compensatory attempt to sustain energy production under toxic conditions. This heightened oxygen consumption rate could also predispose the developing vasculature to long-term dysfunction. Future studies will focus on determining the mechanism(s) underlying this ethanol-driven impairment of mitochondrial function during cerebral artery development and its impact on the neurodevelopmental delays observed in FASD.

## Figures and Tables

**Figure 1 biomolecules-15-01566-f001:**
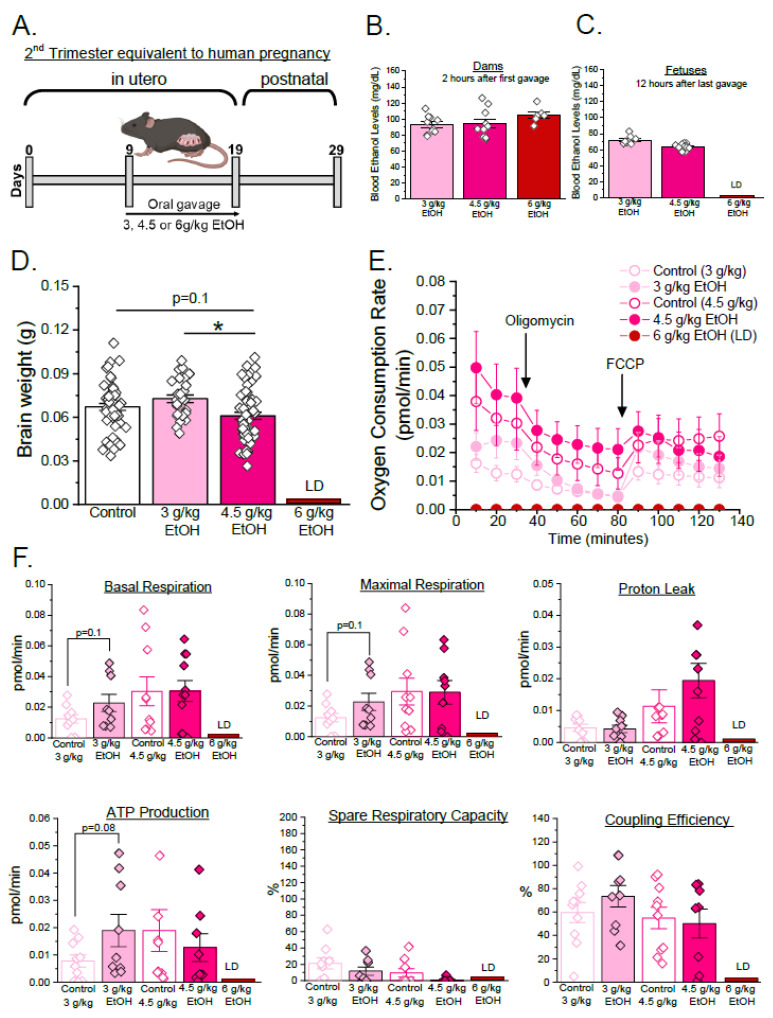
Concentration-specific effects of ethanol during the second trimester equivalent of human pregnancy in a mouse model. (**A**) Experimental design: Pregnant C57BL/6J mice were randomized into several experimental groups based on varying concentrations of ethanol (3, 4.5, or 6 g/kg) via oral gavage, twice daily, from GD 9 to 19. The corresponding control groups were receiving distilled water via gavage in the amount that matched their respective ethanol-exposed counterparts. (**B**) Maternal blood ethanol levels were measured 2 h after first ethanol exposure (3 g/kg EtOH *n* = 10, 4 g/kg EtOH *n* = 10, 6 g/kg EtOH *n* = 6). (**C**) Fetal blood ethanol levels were measured 12 h post final ethanol exposure (3 g/kg maternal EtOH *n* = 9, 4.5 g/kg maternal EtOH *n* = 15; LD = lethal dose). Here, and in all other panels and figures, data are shown as mean ± SEM. LD: lethal dose. (**D**) Fetal brain weights measured at GD 19 (control *n* = 52, 3 g/kg maternal EtOH *n* = 26, 4.5 g/kg maternal EtOH *n* = 57). Each point represents an individual fetus. * *p* < 0.05, one-way ANOVA with Tukey’s post hoc test. (**E**) Seahorse Mito Stress Test assay of isolated fetal cerebral arteries showing oxygen consumption rate (OCR). In this plot, non-mitochondrial oxygen consumption obtained after addition of rotenone/antimycin mixture was subtracted from each datapoint to only render oxygen consumption values associated with mitochondrial respiration. (**F**) Mitochondrial respiratory parameters (basal respiration, maximal respiration, proton leak, ATP production-associated oxygen consumption, spare respiratory capacity, and coupling efficiency). Each point (“*n*”) represents data collected from a pooled artery tissue of six fetuses from a single dam. Data were collected from the following numbers of dams: control 3 g/kg, *n* = 10; 3 g/kg maternal EtOH, *n* = 9; control 4.5 g/kg, *n* = 10; and 4.5 g/kg maternal EtOH, *n* = 9. Data are shown as mean ± SEM. *p*-values represent statistical trends (0.05 < *p* ≤ 0.1) by one-way ANOVA with Tukey’s post hoc test.

**Figure 2 biomolecules-15-01566-f002:**
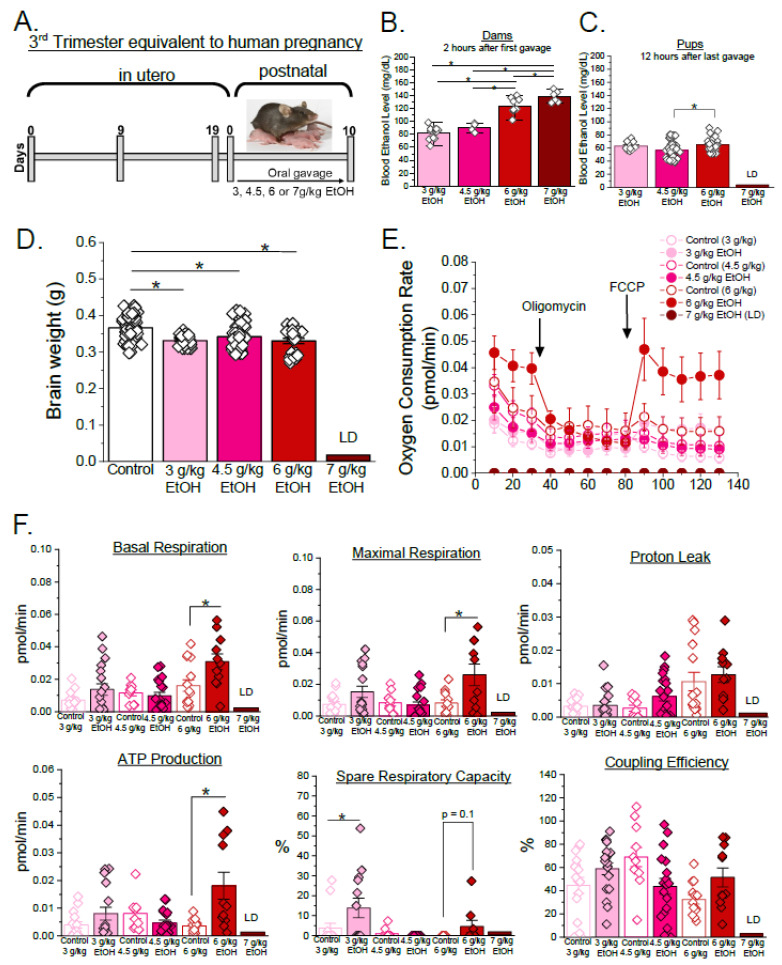
Concentration-specific effects of ethanol during the third trimester equivalent of human pregnancy in a mouse model. (**A**) Experimental design: Nursing C57BL/6J mice were randomized to receive varying concentrations of ethanol (3, 4.5, 6, or 7 g/kg) via oral gavage, twice daily, from PD 1 to 10. Control groups were receiving distilled water via gavage in the amount that was equivalent to the total volume of gavage in the corresponding ethanol-exposed group. (**B**) Maternal blood ethanol levels were measured 2 h after the first gavage (3 g/kg EtOH *n* = 14, 4.5 g/kg EtOH *n* = 4, 6 g/kg EtOH *n* = 9, 7 g/kg EtOH *n* = 6). Here, and in (**C**,**D**,**F**), * *p* < 0.05 by one-way ANOVA with Tukey’s post hoc test. (**C**) Pup blood ethanol levels were measured 12 h after the mother received the last gavage (3 g/kg maternal EtOH *n* = 10, 4.5 g/kg maternal EtOH *n* = 54, 6 g/kg maternal EtOH *n* = 49). (**D**) Fetal brain weights measured at PD 11. Each point represents an individual pup (control *n* = 79, 3 g/kg maternal EtOH = 29, 4.5 g/kg maternal EtOH *n* = 95, 6 g/kg maternal EtOH *n* = 24). Here, and in (**E**,**F**), data were obtained from litters of no less than 3 dams. (**E**) Seahorse Mito Stress Test assay of isolated fetal cerebral arteries showing oxygen consumption rate (OCR). In this plot, non-mitochondrial oxygen consumption obtained after addition of rotenone/antimycin mixture was subtracted from each datapoint to only render oxygen consumption values associated with mitochondrial respiration. (**F**) Mitochondrial respiratory parameters (basal respiration, maximal respiration, proton leak, ATP production-associated oxygen consumption, spare respiratory capacity, and coupling efficiency). Each point represents data collected from a pooled artery of three pups. No more than two datapoints were obtained from a single dam. Data were collected from the following numbers of dams: control 3 g/kg *n* = 9, 3 g/kg maternal EtOH *n* = 12, control 4.5 g/kg *n* = 11, 4.5 g/kg maternal EtOH *n* = 11, control 6 g/kg *n* = 8, 6 g/kg maternal EtOH *n* = 7. Data are shown as mean ± SEM. * *p* < 0.05 by one-way ANOVA with Tukey’s post hoc test. *p*-values represent statistical trends (0.05 < *p* ≤ 0.1) by one-way ANOVA with Tukey’s post hoc test.

**Figure 3 biomolecules-15-01566-f003:**
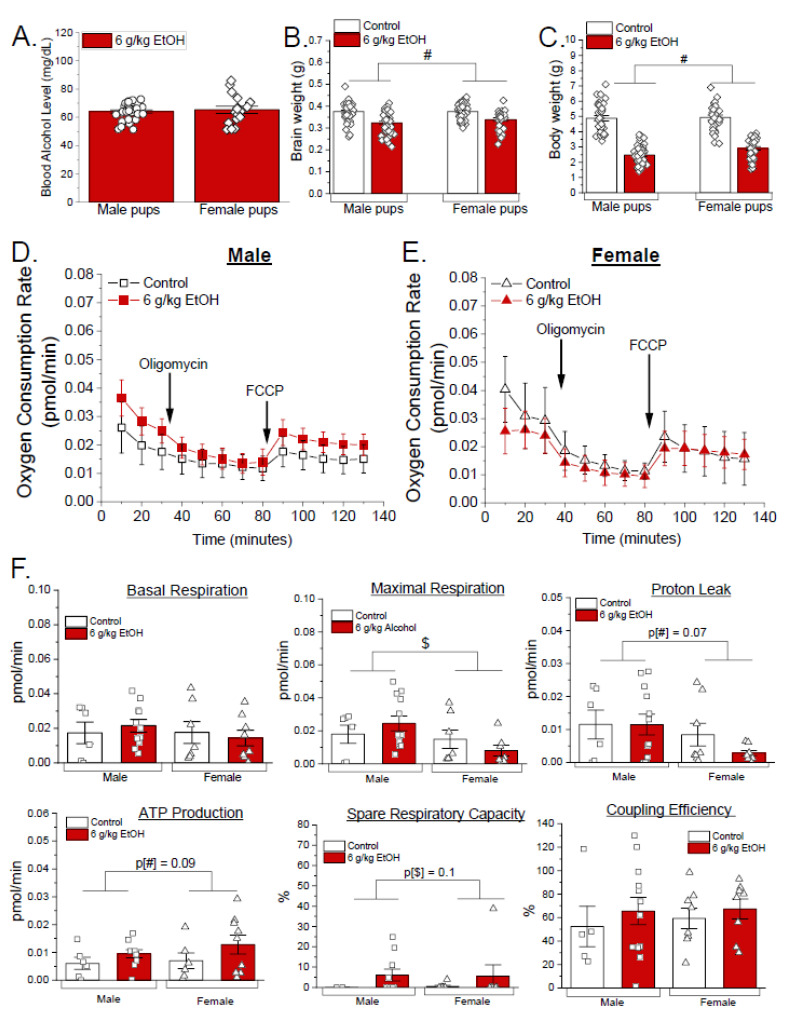
Sex-specific effects of ethanol exposure on oxygen consumption and mitochondrial respiration parameters in developing fetal cerebral arteries. (**A**) Male versus female pup blood ethanol level 12 h after the dam received ethanol gavage (6 g/kg ethanol). Control dams were receiving distilled water in the volume that was equal to ethanol-containing gavages. Male pups are represented by hollow squares, and female pups are represented by hollow triangles. Each point represents an individual fetus (male *n* = 31, female *n* = 19). *p* = 0.66 by two-tailed Welch’s *t*-test. (**B**) Brain weights of the male and female pups. Each point represents an individual pup (male control *n* = 48, male 6 g/kg maternal EtOH *n* = 43, female control *n* = 48, female 6 g/kg maternal EtOH *n* = 30). # *p* < 0.05 by two-way ANOVA, factor: ethanol exposure. (**C**) Body weights of the male and female pups. Each point represents an individual pup (male control *n* = 34, male 6 g/kg maternal EtOH *n* = 45, female control *n* = 29, female 6 g/kg maternal EtOH *n* = 34). Seahorse Mito Stress Test assay of the isolated male (**D**) and female (**E**) cerebral arteries shows oxygen consumption rate. In these plots, non-mitochondrial oxygen consumption obtained after addition of rotenone–antimycin mixture was subtracted from each datapoint to only render oxygen consumption values associated with mitochondrial respiration. (**F**) Mitochondrial respiration parameters (basal respiration, maximal respiration, proton leak, ATP production-associated oxygen consumption, spare respiratory capacity, and coupling efficiency) were measured. Each point represents data collected from a pooled artery of three pups. No more than two male and two female data points were obtained from a single dam. Number of dams used is as follows: male control *n* = 6, male 6 g/kg maternal EtOH *n* = 11, female control *n* = 8, female 6 g/kg maternal EtOH *n* = 9. Data is shown as mean ± SEM. # *p* < 0.05 by two-way ANOVA (factor: ethanol exposure); $ *p* < 0.05 by two-way ANOVA (factor: sex).

**Figure 4 biomolecules-15-01566-f004:**
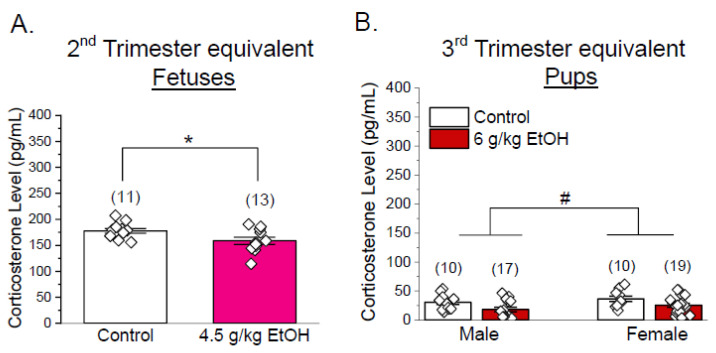
Corticosterone levels in dams receiving ethanol and their offspring. (**A**) Fetal corticosterone levels were measured at GD 19 (control vs. 4.5 g/kg maternal ethanol) in serum. Each point represents an individual dam or pup. Corticosterone levels were measured by an ELISA kit. Data are shown as mean ± SEM. * *p* < 0.05 vs. control, two-tailed Welch’s *t*-test. (**B**) Male and female pup corticosterone levels at PD 10 (control vs. 6 g/kg maternal ethanol exposure during PD1 to 10). Data shown as mean ± SEM. # *p* < 0.05 by two-way ANOVA (factor: ethanol exposure).

**Table 1 biomolecules-15-01566-t001:** Equations used to calculate different mitochondria-associated respiratory parameters derived from Seahorse XF Mito Stress Test assay.

Parameters	Equation
Non-mitochondrial Respiration	Minimum OCR measured throughout
Basal Respiration	(Last OCR measured before oligomycin injection)—(Non-Mitochondrial Respiration)
Maximal OCR (Maximal Respiration)	(Maximum rate measured after FCCP injection)—(Non-Mitochondrial Respiration)
Proton (H^+^) Leak	(Minimum rate measured after oligomycin injection)—(Non-Mitochondrial Respiration)
OCR associated with ATP Production	(Last rate measured before oligomycin injection)—(Minimum rate measured after oligomycin injection)
Spare Respiratory Capacity (%)	(Maximal Respiration-Basal Respiration)/(Basal Respiration) × 100
Coupling Efficiency (%)	(OCR associated with ATP Production)/(Basal Respiration) × 100

## Data Availability

The raw data supporting the conclusions of this article will be made available by the authors on request.

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
