# Peer review of "Ethanol Exposure Increases Oxygen Consumption by Developing Cerebral Arteries in a Trimester-, Concentration- and Sex-Dependent Manner"

_biomolecules, 2025, doi:10.3390/biom15111566_

Round 1
Reviewer 1 Report
Comments and Suggestions for Authors
The study by Thapa et al. titled “Ethanol exposure increases oxygen consumption by developing cerebral arteries in a trimester-, concentration- and sex-dependent manner” examined the effect of prenatal and postnatal alcohol exposures, modeling 2nd and 3rd trimester equivalent exposures, respectively, on various measures of mitochondrial function from cerebral arteries. Overall, it was found that prenatal alcohol exposed offspring showed subtle alterations in mitochondrial function that were timing of exposure-, dose-, and sex-specific. This study provides novel evidence that further supports the complexity of prenatal alcohol exposure effects, and highlight the sensitivity of metabolic processes, specifically mitochondrial, within the cerebral vasculature to alcohol during critical developmental periods. Below are some specific comments that should be considered to improve the interpretations of the findings:
- A reference needs to be provided for why PD 1-10 is the equivalent to the 3rd trimester.
- Has the lab previously measured BECs from pups at earlier time points post ethanol administration using this PAE model in order to provide an estimate of how high the BECs may have gotten? If so, this should be referenced.
- While it makes sense why corticosterone levels were measured from blood that was taken at after thoracotomy, what was the purpose of measuring CORT at this time point?
- Related to the previous comment, CORT levels were measured hours after the final ethanol exposure/injection but at a single time only, capturing a snapshot of the dynamic HPA axis response. Thus, the conclusion that CORT does not contribute to the observed PEE effects should be better discussed.
- In the methods, a clearer description about which animal (nursing dam or pup) receives ethanol administration in the 3rd trimester-equivalent model needs to be provided. The way it is currently written, I was confused until I read the figure 2 caption.
- If available, data on litter sizes should be presented for the gestational exposure group.
- There is concern regarding the report of "trends" (although defined based on the p-value), since there are several trends that are clearly a result of 1 or 2 data points that are skewing the mean, resulting in a suggestive pattern of effect (i.e. Figure 3F - spare respiratory capacity). The conclusions derived from these trends should be tempered in the results and discussion. Additionally, it may be more beneficial to not report the trends since the variability of some of the data is quite high.
no comment
Author Response
We deeply thank you for your time, effort, and constructive criticism. We fully addressed your comments, questions, and suggestions as specified in the attached document.

Reviewer 2 Report
Comments and Suggestions for Authors
Major Comments
-There is no correlation with markers of oxidative stress (ROS, 4-HNE, SOD2, mtDNA damage), respiratory chain proteins, or markers of apoptosis/angiogenesis. Assessing oxidative stress would help substantiate the argument for mitochondrial dysfunction. Including quantification of mitochondrial proteins and oxidative stress markers is essential for the interpretation and discussion of the data.
-The use of simple ANOVA without correction for multiple testing may increase the risk of false positives. Some figures report “trends” (0.05 < p ≤ 0.1) as biologically relevant, which weakens inferential rigor. Small sample sizes for OCR (n = 3–4 dams per group) limit statistical power. The number of dams per group should be expanded or independent replications performed. Mixed-model analysis considering “dam” as a random factor is recommended. Some trends (p = 0.08–0.1) are interpreted as nearly significant, which can be problematic.
-Maternal mortality data (50% at 6 g/kg) suggest excessive toxicity, possibly inadequate for biological comparison with human doses. The authors should discuss more clearly how the 3–6 g/kg levels in mice relate to human binge-drinking patterns.
-The increase in OCR is interpreted as compensatory, but it could instead reflect respiratory leak or mitochondrial uncoupling. Supplementary Fig. 1 does not provide quantitative proton leak values sufficient to rule out this hypothesis.
-The study does not discuss potential contributions of the placenta in regulating ethanol and oxygen transfer. Why was the placenta not considered as a key modulator of oxygen and ethanol transport? Similarly, the relationship between elevated OCR and structural damage is not explored; for instance, whether the increase is adaptive or pre-pathological.
-The discussion of sex differences remains superficial: it does not integrate literature on mitochondrial modulation by estrogen/testosterone or on fetal microglia. Corticosterone was analyzed in isolation; other endocrine axes (ACTH, aldosterone, estradiol/testosterone) could clarify sex differences.
-Overall, the measurements are purely biochemical, lacking functional data (vasoreactivity, arterial structure, mitochondrial density). The “metabolic compensation” hypothesis is plausible but not directly tested. Discussion statements should be cautious and softened throughout the manuscript.
Minor Comments
-The introduction should highlight the clinical relevance for preventive interventions in FASD and also consider the role of ethanol metabolism (acetaldehyde, ROS) and fetal hypoxia. Update the introduction by including more recent references on mitochondrial metabolism in FASD.
-Correct minor inconsistencies in figure formatting and units (mg/dL vs mM).
Author Response

(The authors gave the same response as above.)
